# HE-Drive: Human-Like End-to-End Driving with Vision Language Models

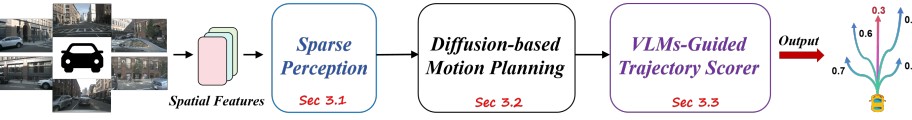

(a) Our End-to-End Autonomous Driving Paradigm

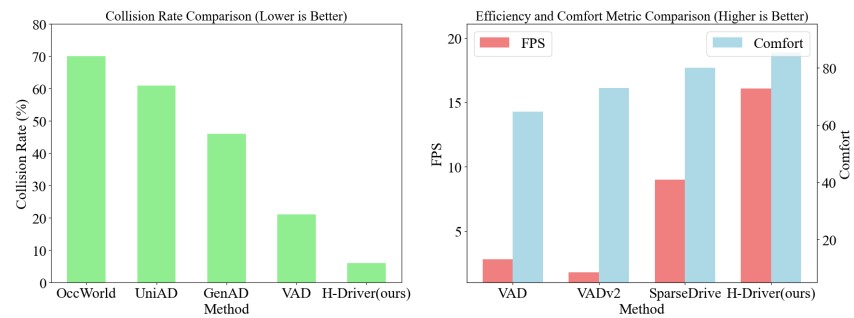

(b) Comparison of Performance (i.e. collision rate), Efficiency (i.e. FPS), and Comfort Metrics

Figure 1: We present HE-Drive, the first Human-like end-to-end driving system. HE-Drive takes the multi-view sensor data as input and outputs the optimal path to drive in complex scenarios.

## ABSTRACT

In this paper, we propose **HE-Drive**: the first human-like-centric end-to-end autonomous driving system to generate trajectories that are both temporally consistent and comfortable. Recent studies have shown that imitation learning-based planners and learning-based trajectory scorers can effectively generate and select accuracy trajectories that closely mimic expert demonstrations. However, such trajectory planners and scorers face the dilemma of generating *temporally inconsistent* and *uncomfortable* trajectories. To solve the above problems, Our HE-Drive first extracts key 3D spatial representations through sparse perception, which then serves as conditional inputs for the Conditional Denoising Diffusion Probabilistic Model (DDPM)-based motion planner to generate temporal consistency multi-modal trajectories. A Vision-Language Model (VLM)-guided trajectory scorer subsequently selects the most comfortable trajectory from these candidates to control the vehicle, ensuring human-like end-to-end driving. Experiments show that HE-Drive not only achieves state-of-the-art performance (*i.e., reduces the average collision rate by 71% than VAD*) and efficiency (*i.e., 1.9× faster than SparseDrive*) on the challenging nuScenes and OpenScene datasets but also provides the most comfortable driving experience on real-world data.

## 1 INTRODUCTION

The end-to-end paradigm (Hu et al., 2023b; Jiang et al., 2023b; Sun et al., 2024), which integrates perception, planning and trajectory scoring tasks into a unified model optimized for planning objectives, has recently demonstrated significant potential in advancing autonomous driving technology (Figure 1a). The latest research proposes imitation learning-based motion planners (Chen et al.,

2024; Cheng et al., 2024) that learn driving strategies from large-scale driving demonstrations and employ learning-based trajectory scorers (Zhao et al., 2021; Jiang et al., 2023a) to select the safest and most accurate trajectory from multiple predicted candidates to control the vehicle. However, despite the significant improvements in prediction accuracy achieved by existing planners and scorers, they face the challenges of generating *temporally inconsistent* trajectories, where consecutive predictions are unstable and inconsistent over time, and selecting *uncomfortable* trajectories that exhibit continuous braking, resulting in stalling or excessive turning curvature.

In this work, we introduce **HE-Drive**, the first human-like-centric end-to-end autonomous driving system to solve the above two problems, as illustrated in Figure 2. Specifically, we find that the *temporal inconsistency* in trajectories generated by imitation learning-based planners arises from two main factors: temporal correlation and generalization. Firstly, these planners rely on the past few seconds of information from the current frame to forecast future trajectories, ignoring the correlation between consecutive predictions (Zhou et al., 2023; Tang et al., 2024). Secondly, their performance is constrained by the quality of the collected offline expert trajectories, rendering them susceptible to changes in system dynamics and out-of-distribution states, resulting in learned policies that lack generalization ability to unseen scenarios. Inspired by the success of the diffusion policy (Chi et al., 2024) in robotic manipulation, which employs a vision-conditioned diffusion model (Ho et al., 2020) to accurately represent multi-modal distributions for generating action sequences, we propose a diffusion-based planner that generates multi-modal trajectories with strong temporal consistency.

Furthermore, the key reason for the *uncomfortable* predicted trajectories is the suboptimal trajectory scorer's inability to achieve lifelong evaluation and the absence of universal metrics to measure trajectory comfort. Recent studies have revealed that learning-based scorers are inferior to rule-based scorers in closed-loop scenarios (Dauner et al., 2023), while the latter suffers from limited generalization due to their reliance on hand-crafted post-processing. Other researchers have explored the use of Vision-Language Models (VLMs) (Shao et al., 2024; Sima et al., 2023; Xu et al., 2024a) to perceive the motion and traffic representation of surrounding agents to decide the next movement. However, directly employing VLMs as driving decision-makers poses challenges related to poor interpretability and severe hallucinations (Xu et al., 2024b). To address these issues, we propose a novel trajectory scorer and universal comfort metric that combines the interpretability of rule-based scorers with the adaptability of VLMs to adjust driving styles (*i.e., aggressive or conservative*) for lifelong evaluation.

In summary, HE-Drive, the novel human-like-centric end-to-end autonomous driving system, utilizes sparse perception to detect, track, and map driving scenarios based on sparse features, generating 3D spatial representations. These representations are conditionally input into the diffusion-based motion planner, powered by the Conditional Denoising Diffusion Probabilistic Model (DDPM). Finally, a VLM-guided (*i.e., Llama 3.2V*) trajectory scorer selects the most comfortable trajectory from the candidates to control the vehicle, ensuring human-like end-to-end driving. The main contributions of our work are summarized as follows:

- **Diffusion-based Motion Planner:** We propose a diffusion-based motion planner that generates temporal consistent and multi-modal trajectories by conditioning on the 3D representation extracted by the sparse perception network and incorporating the speed, acceleration, and yaw of the historical prediction trajectory. (§ 3.2)

- **Plug-and-Play Trajectory Scorer:** We introduce a novel VLMs-guided trajectory scorer and a comfort metric, which address the gap in human-like driving, making it easily integrable into existing autonomous driving systems. (§ 3.3)

- **Excellent Results in Open-loop and Closed-loop Benchmarks:** HE-Drive achieves state-of-the-art performance (*i.e.*, reduces the average collision rate by **71%** compared to VAD) and efficiency (*i.e.*, **1.9×** faster than SparseDrive) on nuScenes and OpenScene datasets, while increasing comfort by **32%** on real-world datasets, showcasing its effectiveness across various scenarios. (§ 4.2, § 4.4, and § 4.5)

## 2 RELATED WORK

In this section, we first review classical approaches to end-to-end autonomous vehicle navigation in Section 2.1. Following that, Section 2.2 aggregates the current research on employing diffusion

models for trajectory planning in robotics. Advancing our discussion, Section 2.3 reviews the use of VLMs for trajectory evaluation in autonomous driving systems.

## 2.1 END-TO-END AUTONOMOUS DRIVING

End-to-end autonomous driving aims to generate planning trajectories directly from raw sensors. In the field, advancements have been categorized based on their evaluation methods: open-loop and closed-loop systems. In open-loop systems, UniAD (Hu et al., 2023a) presents a unified framework that integrates full-stack driving tasks with query-unified interfaces for improved interaction between tasks. VAD (Jiang et al., 2023a) boosts planning safety and efficiency, evidenced by its performance on the nuScenes dataset, while SparseDrive (Sun et al., 2024) utilizes sparse representations to mitigate information loss and error propagation inherent in modular systems, enhancing both task performance and computational efficiency. For closed-loop evaluations, VADv2 (Chen et al., 2024) advances vectorized autonomous driving with probabilistic planning, using multi-view images to generate action distributions for vehicle control, excelling in the CARLA Town05 benchmark.

## 2.2 DIFFUSION MODELS FOR TRAJECTORY GENERATION

Diffusion models initially celebrated in image synthesis, have been adeptly repurposed for trajectory generation. Potential-Based Diffusion Motion Planning (Luo et al., 2024) further enhances the field by employing learned potential functions to construct adaptable motion plans for cluttered environments, demonstrating the method's scalability and transferability. NoMaD (Sridhar et al., 2024) and SkillDiffuser (Liang et al., 2024) both present unified frameworks that streamline goal-oriented navigation and skill-based task execution, respectively, with NoMaD achieving improved navigation outcomes and SkillDiffuser enabling interpretable, high-level instruction following. In a word, diffusion models offer a promising alternative to imitation learning-based end-to-end autonomous driving frameworks for trajectory planning. Imitation learning models may incorrectly attribute a driver's actions to the wrong causal factors due to inherent causal confusion. In contrast, diffusion models can better capture the underlying causal relationships by learning the joint distribution of scene features and driver actions in the latent space, enabling the model to correctly associate the true causes with the appropriate actions.

## 2.3 LARGE LANGUAGE MODELS (LLMS) FOR TRAJECTORY EVALUATION

Trajectory scoring (Fan et al., 2018) plays a vital role in autonomous driving decision-making. Rule-based methods (Treiber et al., 2000) provide strong safety guarantees but lack flexibility, while learning-based methods (Chitta et al., 2021; Prakash et al., 2021) perform well in open-loop tasks but struggle in closed-loop scenarios (Treiber et al., 2000; Dauner et al., 2023). Recently, DriveLM (Sima et al., 2023) integrates VLMs into end-to-end driving systems, modelling graph-structured reasoning through perception, prediction, and planning question-answer pairs. However, the generated results of large models may contain hallucinations and require further strategies for safe application in autonomous driving. The emergence of VLMs raises the question: *Can VLMs adaptively adjust driving style while ensuring comfort based on a trajectory scorer?*

## 3 METHODOLOGY

In this section, we introduce the key components of **HE-Drive** (Figure 2): sparse perception (Sec 3.1), diffusion-based motion planner (Sec 3.2), and trajectory scorer guided by VLMs (Sec 3.3).

## 3.1 SPARSE PERCEPTION

HE-Drive begins by employing a visual encoder (He et al., 2016) to extract multi-view visual features, denoted as $\mathcal{F}$, from the input multi-view camera images. These images denoted as $\Gamma = \{J_\tau \in \mathbb{R}^{N \times 3 \times H \times W}\}_{\tau=T-k}^{T}$, where $N$ is the number of camera views, $k$ is the temporal window length, and $J_\tau$ represents the multi-view images at timestep $\tau$, with $T$ being the current timestep. Subsequently, the sparse perception from (Sun et al., 2024) performs detection, tracking, and online mapping tasks concurrently offering a more efficient and compact 3D representation $\Theta$ of the surrounding environment (in Figure 2).

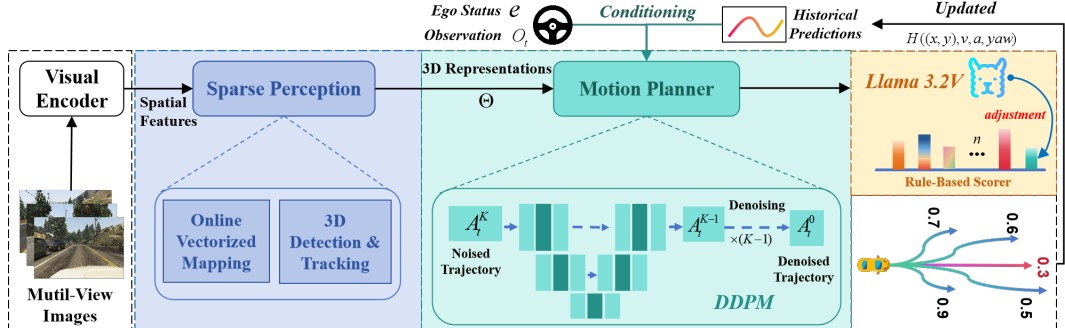

Figure 2: **Overview of our proposed framework.** HE-Drive first extracts features from multi-view images using an off-the-shelf visual encoder then perceives dynamic and static elements sparsely to generate 3D representation. The above representations and historical prediction trajectories are used as conditions of the diffusion model to generate *temporal consistency* multi-modal trajectories. The final trajectory scorer selects the most *comfortable* trajectory from these candidates to control the vehicle.

## 3.2 DIFFUSION-BASED MOTION PLANNER

Figure 2 illustrates the overall pipeline of our diffusion-based motion planner. We adopt the CNN-based diffusion policy (Chi et al., 2024; Ze et al., 2024) as the foundation, which consists of a conditional U-Net composed of 1D convolution layers, upsampling layers, and FiLM (Feature-wise Linear Modulation) layers (Perez et al., 2018).

**Motion Planner Diffusion Policy:** Our method (Figure 7) employs the Conditional Denoising Diffusion Probabilistic Model (DDPM), a generative model defined through parameterized Markov chains trained using variational inference to model the conditional distribution $p(\mathbf{A}_t|\mathbf{O}_t)$. The DDPM consists of a forward process that gradually adds Gaussian noise to the input data, converting it into pure noise, and a reverse process that iteratively denoises the noisy data to recover the original data.

Specifically, the input conditions to the DDPM include the compact 3D representation $\Theta$, ego status $e$, historical predicted trajectories $\mathcal{H}$, and their corresponding velocity $v_i$, acceleration $a_i$, and yaw encoding $\theta_i$. The concatenated condition $C$, which includes observation $O_t$ and above relevant conditions, is injected into every convolutional layer of the network using FiLM (Perez et al., 2018). This channel-wise conditioning guides the trajectory generation from the ego position to the anchor position. The denoising process begins with a Gaussian noise $\mathbf{A}_t^k$ of shape [B, $N_a$, $T_i$, P], where B denotes the batch size, $N_a$ represents the number of anchors, $T_i$ indicates the interval time (i = 0.5, 1, 1.5, 2, 2.5, 3) between navigation points on the trajectory. P represents the position (x, y) at each interval time $T_i$. The noisy data is iteratively refined into a noise-free 3s future multi-modal trajectory $\mathbf{A}_0$ through $k$ iterations using the denoising network $\epsilon_\theta$. Each trajectory $\tau_i$ is represented as a set of waypoints $\{(x_t, y_t)\}_{t=1}^{T_i}$. The reverse process is described by the following equation:

$$\mathbf{A}_t^{k-1} = \alpha(\mathbf{A}_t^k - \gamma\epsilon_\theta(\mathbf{A}_t^k, k, \mathbf{O}_t, \Theta, e, \mathcal{H}) + \mathcal{N}(0, \sigma^2, I)) \tag{1}$$

where $\alpha$ and $\gamma$ are scaling factors, and $\mathcal{N}(0, \sigma^2, I)$ represents Gaussian noise with mean 0 and variance $\sigma^2$. Our motion planner leverages the DDPM's ability to generate high-quality samples by iteratively refining the noisy data, conditioned on the relevant input variables. The conditioning information, including the compact 3D representation, ego status, historical trajectories, and their corresponding velocity, acceleration, and yaw encoding, is incorporated into the denoising network through FiLM layers, enabling the generation of multi-modal and strong temporal consistency trajectories that take into account the surrounding environment and historical information. Please refer to the appendix A.1 for a detailed description.

To select the most suitable path from the multi-modal trajectories generated by DDPM, we introduce the VLMs-Guided Trajectory Scorer (VTS), as shown in Figure 3. To our knowledge, VTS is the first trajectory scorer that combines interpretability and zero-shot driving reasoning capabilities. By leveraging Vision-Language Models (VLMs), VTS effectively evaluates trajectories based on various driving factors (e.g., collision probability and comfort), enabling transparent decision-making and adaptability to new driving scenarios without extensive fine-tuning (i.e., lifelong evaluation).

Table 1: Weights of Rule-based Scorers

| Cost Category | Specific Cost | Weight |
|---|---|---|
| Safety | $w_{coll}$ | 5.0 |
| Safety | $w_{deviation}$ | 3.5 |
| Safety | $w_{dis}$ | 1.5 |
| Safety | $w_{speed}$ | 2.5 |
| Comfort | $w_{lat}$ | 1.5 |
| Comfort | $w_{lon}$ | 4.5 |
| Comfort | $w_{cent}$ | 3.0 |

### 3.3 VLMs-Guided Trajectory Scorer

#### 3.3.1 Rule-based trajectory scoring strategy

Specifically, We use a linear combination of the following cost functions to score the sampled trajectories. The total cost function, $C_{\text{total}}$, is composed of two main components: safety cost, $C_{\text{safety}}$, and comfort cost, $C_{\text{comfort}}$.

$$C_{\text{total}} = C_{\text{safety}} + C_{\text{comfort}} \quad (2)$$

**Safety Cost:** The safety cost, $C_{\text{safety}}$, is an aggregation of four sub-costs:

$$C_{\text{safety}} = w_{\text{coll}}C_{\text{coll}} + w_{\text{dis}}C_{\text{dis}} + w_{\text{deviation}}C_{\text{deviation}} + w_{\text{speed}}C_{\text{speed}} \quad (3)$$

where

$$C_{\text{coll}} = \exp(-d_{\text{coll}}/\sigma_{\text{coll}}) \quad (4)$$

$$C_{\text{dis}} = \|\mathbf{p}_{\text{end}} - \mathbf{p}_{\text{target}}\|_2 \quad (5)$$

$$C_{\text{deviation}} = \sum_{i=1}^{N}(1 - \cos(\theta_i - \theta_{\text{target}})) \quad (6)$$

$$C_{\text{speed}} = (\bar{v} - v_{\text{target}})^2 \quad (7)$$

Here, $d_{\text{coll}}$ is the minimum distance to obstacles, and $C_{\text{coll}}$ effectively captures the relationship between the vehicle-obstacle distance and the collision risk. The scaling factor $\sigma_{\text{coll}}$ is set to 1.0 meter to ensure a rapid increase in cost as the distance decreases, prioritizing collision avoidance. $\mathbf{p}_{\text{end}}$ and $\mathbf{p}_{\text{target}}$ are the end and target positions, respectively, and $C_{\text{dis}}$ represents the Euclidean distance between them. $N$ is the number of points on the trajectory, $\theta_i$ is the heading of the $i$-th point, and $\theta_{\text{target}}$ is the target heading. $C_{\text{deviation}}$ measures the cumulative deviation of the trajectory from the target heading. $\bar{v}$ is the average speed, $v_{\text{target}}$ is the target speed, and $C_{\text{speed}}$ penalizes deviations from the target speed.

**Comfort Cost:** The comfort cost, $C_{\text{comfort}}$, consists of three sub-costs:

$$C_{\text{comfort}} = w_{\text{lat}}C_{\text{lat}} + w_{\text{lon}}C_{\text{lon}} + w_{\text{cent}}C_{\text{cent}} \quad (8)$$

where

$$C_{\text{lat}} = \max(|a_{\text{lat}}|) \quad (9)$$

$$C_{\text{lon}} = \max(|a_{\text{lon}}|) \quad (10)$$

$$C_{\text{cent}} = \max(|a_{\text{cent}}|) \quad (11)$$

Here, $a_{\text{lat}}$, $a_{\text{lon}}$, and $a_{\text{cent}}$ are the lateral, longitudinal, and centripetal accelerations, respectively. The comfort cost, $C_{\text{comfort}}$, is designed to penalize excessive lateral, longitudinal, and centripetal accelerations that may cause passenger discomfort. By minimizing the maximum absolute values of these accelerations, the trajectory planner aims to reduce sharp side-to-side movements, sudden braking or aggressive acceleration, and ensure smooth navigation through turns. The weights $w_{\text{coll}}$, $w_{\text{dis}}$, $w_{\text{deviation}}$, $w_{\text{speed}}$, $w_{\text{lat}}$, $w_{\text{lon}}$, and $w_{\text{cent}}$ balance the influence of each sub-cost on the overall cost function (in Table 1), allowing the trajectory planner to prioritize different aspects of safety and comfort based on the specific requirements of the autonomous driving system.

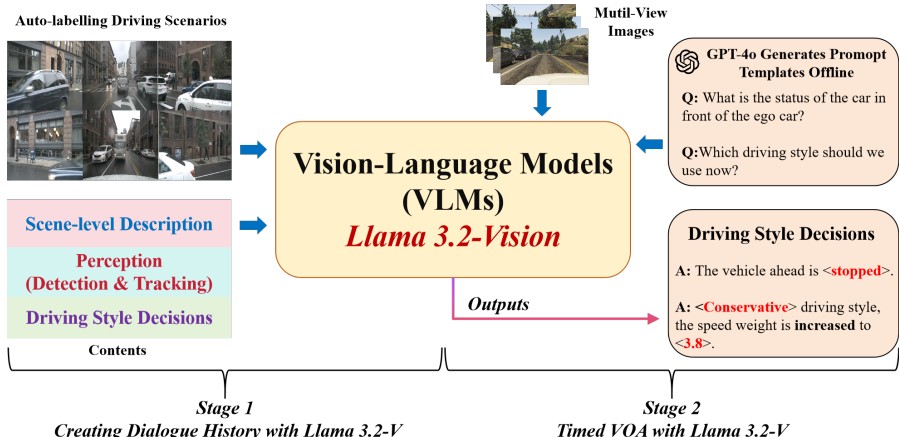

Figure 3: **The overview of the proposed VLMs-guided trajectory scorer (VTS). Stage 1** mitigates hallucination using pre-annotated driving scene data, while **stage 2** conducts VQA with Llama 3.2V for driving style adjustment using GPT-4o generated prompts and vehicle images.

### 3.3.2   VLMs Help Adjust Driving Style.

Our VLM-guided trajectory scorer (Figure 3) consists of two stages. In the first stage, we curate a dataset containing annotated surround images, which include descriptions of the current driving scene, motion states of surrounding agents (i.e., humans and vehicles), and the current driving style (i.e., aggressive or conservative) along with weight adjustment values. Through iterative dialogues, Llama 3.2V (Dubey et al., 2024) assimilates contextual information, mitigating model hallucinations.

In the second stage, we generate a series of prompt templates using GPT-4o (Achiam et al., 2023) for visual question answering (VQA). Leveraging the spatial-temporal stability inherent in traffic patterns, we activate Llama 3.2V intermittently every five seconds to refine driving behaviours. The model assesses the driving context when processing new imagery and calibrates the scoring weights for predefined safety and comfort parameters within the rule-based system. This approach allows for precise alterations to cost weights, enhancing the responsiveness of the driving style to varying scenarios. *By employing the VLM as a driving style regulator rather than a direct decision maker, we mitigate safety risks associated with model hallucinations and improve rule-based trajectory scorer adaptability for new scenarios.*

### 3.4   End-to-End Driving Comfort Metric

To address the lack of a universal comfort evaluation metric in existing end-to-end methods, we propose a general metric to assess the comfort and human likeness of predicted trajectories (Han et al., 2023). Our proposed comfort metric aims to quantify the similarity between the predicted trajectory and a ground true trajectory, considering factors such as dynamic feasibility, jerk, and trajectory smoothness.

Considering the simplified kinematic bicycle model in the Cartesian coordinate frame, we describe the dynamics of a front-wheel driven and steered four-wheel vehicle with perfect rolling and no slipping. The state vector is defined as $\mathbf{x} = (p_x, p_y, \theta, v, a_t, a_n, \phi, \kappa)^T$, where $\mathbf{p} = (p_x, p_y)^T$ represents the position at the centre of the rear wheels, $v$ is the longitudinal velocity w.r.t vehicle's body frame, $a_t$ and $a_n$ denote the longitudinal and lateral accelerations, $\phi$ is the steering angle of the front wheels, and $\kappa$ is the curvature. The complete trajectory representation $\sigma(t) : [0, T_s]$ is formulated as:

$$\sigma(t) = \sigma_i(t - \hat{T}_i), \forall i \in \{1, 2, ..., n\}, t \in [\hat{T}_i, \hat{T}_{i+1}),  \tag{12}$$

where $T_s = \sum_{i=1}^{n} T_i$ is the duration of the entire trajectory, and $\hat{T}_i = \sum_{j=1}^{i-1} T_j$ is the timestamp of the starting point of the $i$-th segment, with $\hat{T}_1 = 0$. The comfort metric is defined as:

$$C = \sum_{k=1}^{3} \int_{0}^{T_k} (w_1|a_t - a_t^*| + w_2|a_n - a_n^*| + w_3|\dot{\phi} - \dot{\phi}^*| + w_4|j_t - j_t^*| + w_5|j_n - j_n^*| + w_6|\dot{\kappa} - \dot{\kappa}^*|)dt, \tag{13}$$

where $T_k \in \{1s, 2s, 3s\}$ represents the considered trajectory duration, $a_t^*, a_n^*, \dot{\phi}^*, j_t^*, j_n^*$, and $\dot{\kappa}^*$ are the corresponding values from the ground true trajectory, and $w_1, w_2, w_3, w_4, w_5, w_6$ are weighting factors for longitudinal acceleration, lateral acceleration, steering angle rate, longitudinal jerk, lateral jerk, and curvature rate, respectively. The longitudinal and lateral jerk, $j_t$ and $j_n$ are calculated as the time derivatives of $a_t$ and $a_n$, respectively.

By calculating the difference between the predicted trajectory and the ground true trajectory for each of these aspects and summing the differences for each time horizon, we obtain an overall difference score. A lower score indicates a higher level of comfort and similarity to the expert trajectory. Finally, by introducing a normalization factor, we expressed the comfort index as a percentage for easier comparative analysis. Additional details can be found in the Appendix A.2.

## 4 EXPERIMENTS

### 4.1 EXPERIMENT SETUP

**Datasets:** Our experiments are conducted on the 3 challenging datasets, i.e., the nuScenes dataset and the real-world dataset for open-loop testing, complemented by the OpenScene dataset for closed-loop evaluation. Specifically, nuScenes dataset (Caesar et al., 2020), encompasses 1,000 detailed driving scenes, each with a duration of approximately 20 seconds. OpenScene dataset (Contributors, 2023; Dauner et al., 2024) is specifically curated to mitigate the prevalent issues found in open-loop evaluation protocols by employing navigation goals extracted from lane graphs and introducing simulation-based metrics that serve as a more robust alternative to displacement errors.

**Implementation Details:** The training process of HE-Drive is divided into multiple stages. Firstly, we train the sparse perception component following the two-stage approach proposed in SparseDrive (Sun et al., 2024), according to the different perception backbone networks, it is divided into HE-Drive-S and HE-Drive-B. The output of the second stage of the sparse perception training serves as the input to the motion planner. Our motion planner employs a convolutional network-based diffusion policy (Chi et al., 2024) to generate accurate and temporally consistent trajectories. Finally, we perform end-to-end training of the entire HE-Drive system. This end-to-end training is conducted on 8 NVIDIA RTX 4090 GPUs, utilizing the AdamW (Loshchilov, 2017) optimizer with a weight decay of 0.01 and an initial learning rate of 5e-4.

### 4.2 END-TO-END PLANNING RESULTS ON THE NUSCENES

As presented in Table 2, the HE-Drive model demonstrates superior performance and efficiency relative to previous approaches encompassing Camera-based and LiDAR-based methodologies. The model attains the least L2 error while employing a resource-efficient visual backbone. Specifically, HE-Drive achieves a significant reduction in the mean L2 error—namely, a decrement of 17.8%—compared to UniAD, and concurrently decreases the average collision rates by an impressive 68%. This result comes from the excellent strong temporal consistency of HE-Drive predictions, such as the consecutive *frames 4, 5, and 6* in Figure 9. When enhanced with a more strong visual backbone (Sun et al., 2024) and cutting-edge diffusion policy capabilities, HE-Drive brings the average L2 error and collision rates further down to 0.58 and 0.06, respectively. Furthermore, leveraging ego-centric sparse perception, HE-Drive-S attains remarkable efficiency, operating at 16.1 FPS, which is 1.2x and 2.5x faster than SparseDrive and VAD, respectively, while also achieving the best comfort, with a 39.6% improvement in the 3s comfort level compared to UniAD (in Figure 8).

As depicted in Figure 15, the Llama 3.2V Multi-round vision-language dialogues enable efficient adjudication of driving style, contributing crucially to the ego vehicle's ability to calibrate its driving

Table 2: Planning results on the nuScenes validation dataset. †: Reproduced with official checkpoint.

| Method | Input | Reference | L2(m)↓ | | | | Collision Rate(%)↓ | | | | FPS ↑ |
|---|---|---|---|---|---|---|---|---|---|---|---|
| | | | 1s | 2s | 3s | Avg. | 1s | 2s | 3s | Avg. | |
| IL Ratliff et al. (2006) | LiDAR | ICML 2006 | 0.44 | 1.15 | 2.47 | 1.35 | 0.08 | 0.27 | 1.95 | 0.77 | - |
| FF Hu et al. (2021) | LiDAR | CVPR 2021 | 0.55 | 1.20 | 2.54 | 1.43 | 0.06 | 0.17 | 1.07 | 0.43 | - |
| EO Khurana et al. (2022) | LiDAR | ECCV 2022 | 0.67 | 1.36 | 2.78 | 1.60 | 0.04 | 0.09 | 0.88 | 0.33 | - |
| ST-P3 Hu et al. (2022) | Camera | ECCV 2022 | 1.33 | 2.11 | 2.90 | 2.11 | 0.23 | 0.62 | 1.27 | 0.71 | 1.6 |
| OccNet Tong et al. (2023) | Camera | ICCV 2023 | 1.29 | 2.13 | 2.99 | 2.14 | 0.21 | 0.59 | 1.37 | 0.72 | 2.6 |
| UniAD† Hu et al. (2023b) | Camera | CVPR 2023 | 0.45 | 0.70 | 1.04 | 0.73 | 0.62 | 0.58 | 0.63 | 0.61 | 1.8 |
| VAD† Jiang et al. (2023b) | Camera | ICCV 2023 | 0.41 | 0.70 | 1.05 | 0.72 | 0.03 | 0.19 | 0.43 | 0.21 | 4.5 |
| SparseDrive Sun et al. (2024) | Camera | arXiv 2024 | **0.29** | **0.58** | 0.96 | 0.61 | **0.01** | **0.05** | 0.18 | 0.08 | 9.0 |
| OccWorld-T Zheng et al. (2024a) | Camera | ECCV 2024 | 0.54 | 1.36 | 2.66 | 1.52 | 0.12 | 0.40 | 1.59 | 0.70 | 2.8 |
| OccWorld-S Zheng et al. (2024a) | Camera | ECCV 2024 | 0.67 | 1.69 | 3.13 | 1.83 | 0.19 | 1.28 | 4.59 | 2.02 | 2.8 |
| GenAD Zheng et al. (2024b) | Camera | ECCV 2024 | 0.36 | 0.83 | 1.55 | 0.91 | 0.06 | 0.23 | 1.00 | 0.43 | 6.7 |
| HE-Drive-S (Ours) | Camera | - | 0.31 | **0.58** | **0.93** | **0.60** | **0.01** | **0.05** | **0.16** | **0.07** | **16.1** |
| HE-Drive-B (Ours) | Camera | - | 0.30 | **0.56** | **0.89** | **0.58** | **0.00** | **0.03** | **0.14** | **0.06** | **10.0** |

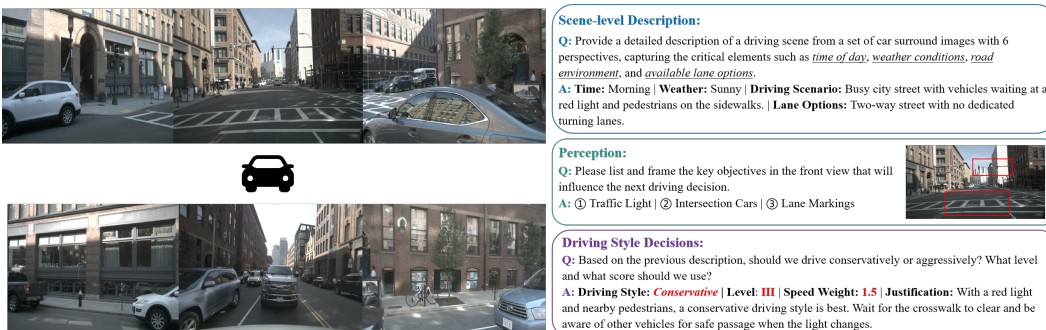

Figure 4: **Qualitative results of Llama 3.2V on nuScenes.** We show the questions (Q), context (C), and answers (A). Incorporating surround view images and textual data, the fine-tuning of driving styles via targeted weight modifications within the rule-based scorer.

approach—whether aggressive or conservative—by taking environmental cues into account, such as perceiving distant traffic signals or appraising the occupancy status of the roadway. This utility is augmented by the VLM's potent zero-shot reasoning capabilities. Subsequent ablation studies are set to further corroborate these findings.

## 4.3 ABLATION STUDY ON THE NUSCENES

We conduct extensive experiments to study the effectiveness and necessity of each design choice proposed in our HE-Drive. We use HE-Drive-S as the default model for ablation.

Table 3: Ablation for designs in parallel motion planner. "HPT" denotes the incorporation of historical prediction trajectory into the DDPM; "H-S&A" represents the inclusion of historical trajectory speed and acceleration in the DDPM; "RTS" refers to the utilization of a rule-based scoring system for planning purposes; "VTS" signifies the employment of a Vision-Language Model (VLM)-guided scoring mechanism.

| ID | HPT | H-S&A | RTS | VTS | Planning L2(m) | | | | Planning Coll.(%) | | | |
|---|---|---|---|---|---|---|---|---|---|---|---|---|
| | | | | | 1s | 2s | 3s | Avg. | 1s | 2s | 3s | Avg. |
| 1 | ✓ | ✓ | ✓ | ✓ | **0.31** | **0.58** | **0.93** | **0.60** | **0.01** | **0.05** | **0.16** | **0.07** |
| 2 | ✓ | ✓ | ✓ | | 0.41 | 0.83 | 1.21 | 0.81 | 0.02 | 0.13 | 0.58 | 0.24 |
| 3 | ✓ | | ✓ | ✓ | 0.34 | 0.67 | 1.11 | 0.71 | 0.03 | 0.09 | 0.20 | 0.11 |
| 4 | | ✓ | ✓ | ✓ | 0.37 | 0.71 | 1.19 | 0.76 | 0.03 | 0.11 | 0.61 | 0.25 |

**Necessity of VLMs:** Incorporating VLMs into the trajectory scoring mechanism primarily aims to facilitate a perpetual evaluative approach in fine-tuning driving behaviours. The absence of VLMs markedly impacts safety metrics, evidenced by a roughly 2.6-fold surge in the 3-second collision rate (Table 3). This phenomenon underscores the limitation of a sole rule-based scorer, which struggles to discern nuanced distinctions across diverse scenarios, thereby complicating trajectory determination.

**Key factors for trajectory consistency:** Enriching the diffusion-based motion planner with historical speed and acceleration data narrows the L2 norm discrepancy observed between the 2s and 3s trajectory predictions. When integrated into the DDPM as conditional variables, these kinetic parameters ensure the generation of coherent trajectories. The coherence hinges not solely on positional coordinates but extends to each navigational point's velocity and acceleration. Moreover, leveraging the temporal correlation of historically optimal predicted trajectories as a conditional element for DDPM proves significantly advantageous, with its omission potentially resulting in a 1.1-fold increment in the L2 norm (Table 3).

Table 4: Ablation study for anchor numbers.

| Number of anchors | Planning L2($m$) | | | | Planning Coll.(%) | | | |
|---|---|---|---|---|---|---|---|---|
| | $1s$ | $2s$ | $3s$ | Avg. | $1s$ | $2s$ | $3s$ | Avg. |
| 2 | 0.36 | 0.72 | 1.19 | 0.76 | 0.03 | 0.11 | 0.60 | 0.25 |
| 4 | 0.36 | 0.65 | 1.08 | 0.70 | 0.01 | 0.09 | 0.45 | 0.18 |
| 6 | 0.35 | 0.62 | 1.03 | 0.66 | 0.00 | 0.05 | 0.37 | 0.14 |
| 8 | 0.31 | 0.58 | 0.93 | 0.60 | 0.00 | 0.03 | 0.14 | 0.06 |
| 10 | 0.34 | 0.63 | 1.07 | 0.68 | 0.01 | 0.05 | 0.43 | 0.16 |

**Necessity of the number of anchors:** We conduct experiments on the number of planning anchors. As shown in Table 4, as the number of planning anchors increases, the planning performance improves continuously until saturated at 8 modes, again proving the importance of multi-modal diffusion planning.

### 4.4 END-TO-END PLANNING RESULTS ON THE REAL-WORLD DATASET

The end-to-end planning results on the real-world dataset are illustrated in the Figure 5a. HE-Drive generates consistent multimodal trajectories and selects the most suitable trajectory with the lowest cost using the trajectory scorer. The purple and green trajectories have higher costs due to their deviation from the target point and reduced comfort during turning manoeuvres.

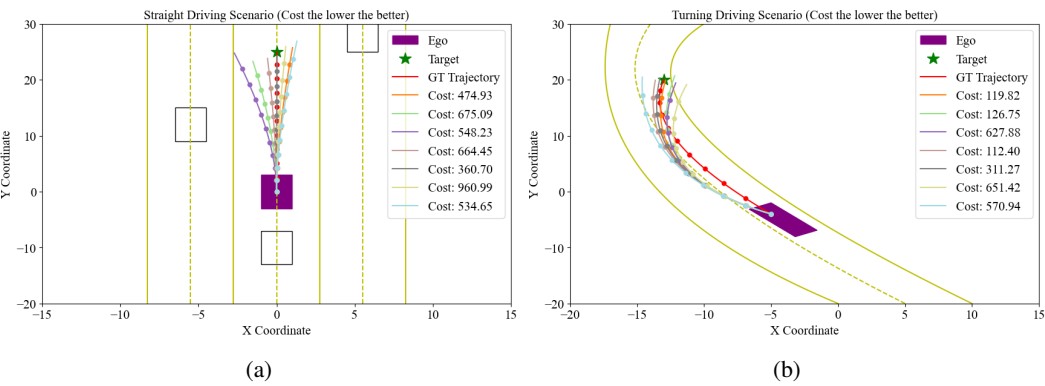

(a)                                                      (b)

Figure 5: (a) and (b) showcase the trajectory generation and scoring process, with the optimal path, indicated by the grey trajectory in (a), being selected for vehicle control based on the lowest cost criterion.

This qualitative result demonstrates that our rule-based scorer prioritizes safety and is interpretable. By adjusting the driving style through VLMs, the most comfortable straight trajectory is selected, as shown in Figure 13 (d), (e), (f). The key reasons are that HE-Drive generates multi-modal trajectories with strong temporal consistency and benefits from the zero-shot generalization ability of VLMs in unseen scenes. Moreover, the comfort metric calculation (Figure 6a) reveals that HE-Drive's 1s

trajectory segment comfort reaches 100%, surpassing VAD by 20%, and the overall 3s trajectory comfort remains higher than VADv2, indicating our scorer's lifelong evaluation capabilities and efficiency for long trajectories.

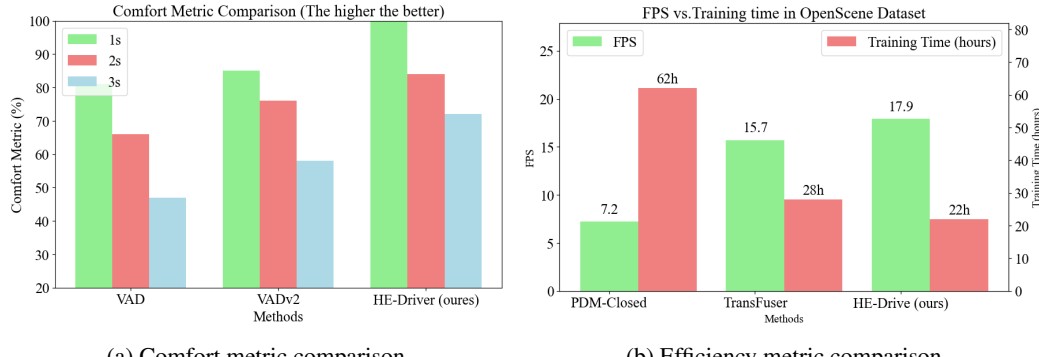

(a) Comfort metric comparison         (b) Efficiency metric comparison

Figure 6: (a) shows the comparison results of HE-Drive and two baselines in terms of the comfort metric in real-world data;(b) shows the comparison results of HE-Drive's efficiency indicators on the closed-loop dataset OpenScene.

### 4.5 END-TO-END PLANNING RESULTS ON THE OPENSCENE DATASET

Our results on the closed-loop dataset: OpenScene presented in Tab. 5 highlight the absolute advantage of HE-Drive over the baseline. In terms of performance, the score is 2.65% higher than that of HyDra-MDP-$\mathcal{V}_{8192}$. In terms of efficiency, HE-Drive demonstrates superior performance compared to its PDM-Closed and TransFuser. It achieves a remarkable 2.56 times higher frames per second (FPS) than PDM-closed, showcasing its exceptional processing speed. Moreover, HE-Drive outpaces TransFuser by 14.01% in FPS (Figure 6b), further highlighting its advanced capabilities. Notably, HE-Drive requires a shorter training period of just 22 hours, making it not only faster in execution but also more efficient in training time. See the Appendix B.4 for more visualization results.

Table 5: **Performance on the OpenScene dataset.** NC, DAC, T T C, C, EP correspond to the No at-fault Collisions, Drivable Area Compliance, Time to Collision, Comfort, and Ego Progress.·

| Method | NC | DAC | EP | TTC | C | Score |
|---|---|---|---|---|---|---|
| Transfuser Chitta et al. (2022) | 96.5 | 87.9 | 73.9 | 90.2 | 100 | 78.0 |
| Vadv2-$\mathcal{V}_{4096}$ Chen et al. (2024) | 97.1 | 88.8 | 74.9 | 91.4 | 100 | 79.7 |
| Vadv2-$\mathcal{V}_{8192}$ Chen et al. (2024) | 97.2 | 89.1 | 76.0 | 91.6 | 100 | 80.9 |
| HyDra-MDP-$\mathcal{V}_{4096}$ Li et al. (2024) | 97.7 | 91.5 | 77.5 | 92.7 | 100 | 82.6 |
| HyDra-MDP-$\mathcal{V}_{8192}$ Li et al. (2024) | 97.9 | 91.7 | 77.6 | 92.9 | 100 | 83.0 |
| HE-Drive (ours) | **98.4** | **92.1** | **78.9** | **96.7** | **100** | **85.2** |

## 5 CONCLUSION

In this paper, we introduce HE-Drive, a novel human-like-centric end-to-end autonomous driving system that addresses the limitations of existing methods in achieving temporal consistency and passenger comfort. HE-Drive integrates a sparse perception module, a diffusion-based motion planner, and a Llama 3.2V guided trajectory scoring system. The sparse perception module achieves a fully sparse scene representation by unifying detection, tracking, and online mapping. The diffusion-based motion planner generates multi-modal trajectories in continuous space, ensuring temporal consistency and mimicking human decision-making. The trajectory scoring module combines rule-based methods with Llama 3.2V to enhance generalization, interpretability, stability, and comfort. Extensive experiments demonstrate HE-Drive's superior performance compared to state-of-the-art methods in both open-loop and closed-loop datasets, generating human-like trajectories with improved temporal consistency and passenger comfort.

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

## A   ARCHITECTURE, TRAINING, AND EVALUATION DETAILS

### A.1   DIFFUSION-BASED PLANNER IMPLEMENTATION DETAILS

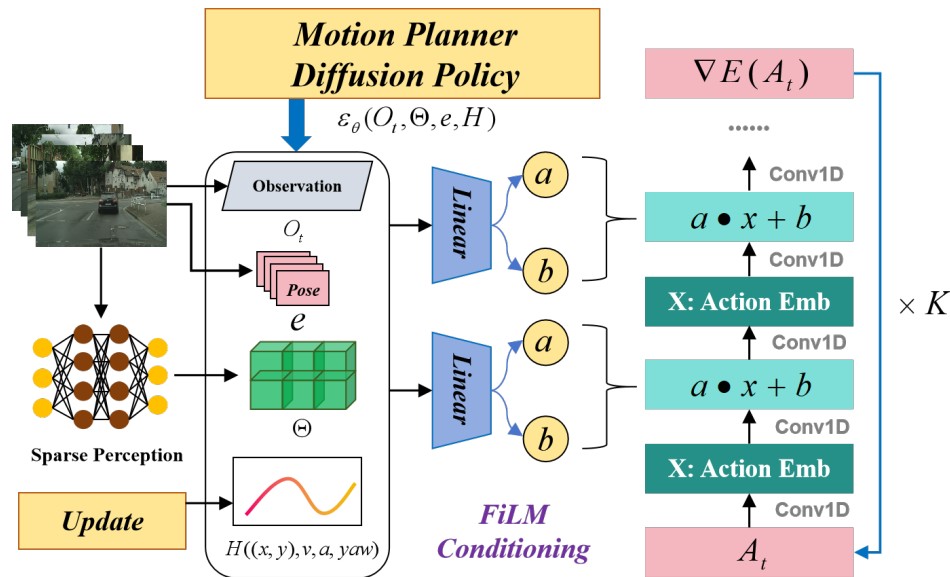

Figure 7: Motion Planner Diffusion Policy Overview. a) General formulation. At time step $t$, the policy takes the latest $T_o$ steps of observation data $\mathbf{O}_t$ as input and outputs $T_a$ steps of trajectories $\mathbf{A}_t$. b) The concatenated condition feature $C$, which includes observation features $\Theta$, ego status $e$, historical predicted trajectories $\mathcal{H}$, and their corresponding velocity $v_i$, acceleration $a_i$, and yaw encoding $\theta_i$, is injected into every convolutional layer using FiLM: $\text{FiLM}(x_i) = \gamma_i \odot x_i + \beta_i$, where $\gamma_i$ and $\beta_i$ are obtained by passing $C$ through a fully connected layer. This channel-wise conditioning guides the trajectory generation from the ego position to the anchor position.

During the training phase, the DDPM learns to estimate the noise component $\epsilon_\theta$ given the noisy input $\mathbf{A}_t^k$, the timestep $k$, and the conditioning variables $\mathbf{O}_t$, $\Theta$, $e$, and $\mathcal{H}$. The conditioning variables are incorporated into the denoising network $\epsilon_\theta$ through the FiLM layers, which modulate the activations of the convolutional layers based on the conditioning information. Specifically, the compact 3D representation $\Theta$ is processed by a separate convolutional neural network to extract relevant features, while the ego status $e$, historical predicted trajectories $\mathcal{H}$, and their corresponding velocity $v_i$, acceleration $a_i$, and yaw encoding $\theta_i$ are concatenated and processed by fully connected layers. The resulting conditioning vectors are then used to modulate the activations of the U-Net's convolutional layers via element-wise affine transformations, enabling the denoising network to adapt its behavior based on the input conditions. By minimizing the difference between the estimated noise and the actual noise added during the forward process, the model learns to denoise the data effectively.

During inference, the model starts with pure Gaussian noise and iteratively denoises it using the learned denoising network $\epsilon_\theta$, conditioned on the input variables, to generate realistic and diverse trajectories. The compact 3D representation $\Theta$ provides the model with essential spatial context, allowing it to understand the environment and generate trajectories that adhere to road constraints and navigate around obstacles. The ego status $e$ and historical predicted trajectories $\mathcal{H}$ enable the model to maintain temporal consistency by considering the vehicle's current state and its previous motion patterns. Furthermore, the velocity $v_i$, acceleration $a_i$, and yaw encoding $\theta_i$ of the historical trajectories provide valuable insights into the vehicle's dynamics and orientation, allowing the DDPMs to generate trajectories that smoothly transition from the past to the future, ensuring realistic and physically plausible motion patterns.

Notably, the Denoising Diffusion Implicit Models (DDIM) approach (Song et al.) can replace DDPM to achieve real-time inference, which decouples the number of denoising iterations in training and inference, thereby allowing the algorithm to use fewer iterations for inference to speed up the diffusion process.

## A.2 COMFORT METRIC

To represent the comfort index as a percentage, we can introduce a normalization factor that scales the comfort index to a value between 0 and 1, with 1 corresponding to a perfect match with the expert trajectory (100% comfort) and 0 indicating the least comfortable trajectory. The normalized comfort index can be expressed as:

$$C_n = e^{-\alpha C}, \tag{14}$$

where $C_n$ is the normalized comfort index, $C$ is the original comfort index as defined in equation (2), and $\alpha$ is a positive scaling factor that determines the sensitivity of the normalized index to changes in the original index. A larger value of $\alpha$ will result in a more rapid decrease in the normalized index as the original index increases.

The comfort percentage can then be calculated as:

$$C_p = 100 \times C_n = 100 \times e^{-\alpha C}. \tag{15}$$

With this formulation, a comfort index of 0 corresponds to a comfort percentage of 100%, while larger values of the comfort index result in lower comfort percentages. The scaling factor $\alpha$ can be tuned based on the desired sensitivity of the comfort percentage to changes in the comfort index. By representing the comfort index as a percentage, we provide a more intuitive understanding of the comfort level of the predicted trajectory relative to the expert trajectory. This normalized representation allows for easier comparison between different trajectories and can be more readily interpreted by users of the end-to-end driving system.

## B ADDITIONAL RESULTS

### B.1 NUSCENES DATASET VISUALIZATION RESULTS

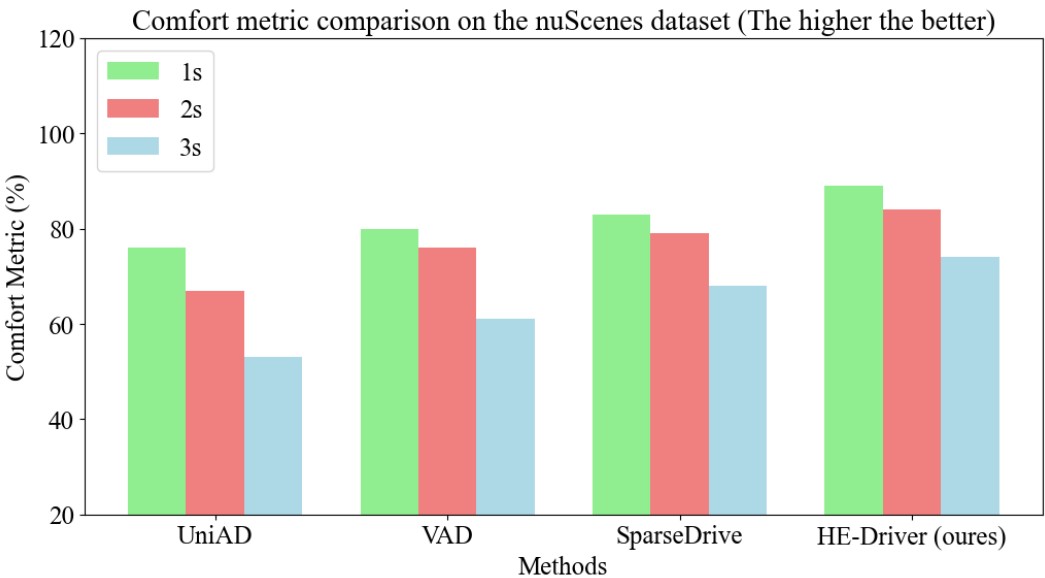

Figure 8: **Quantitative results of comfort metrics on the nuScenes dataset.** HE-Drive achieved the best comfort, with the comfort of 1s improving by 11.3% compared with VAD and the comfort of 3s improving by 39.6% compared with UniAD.

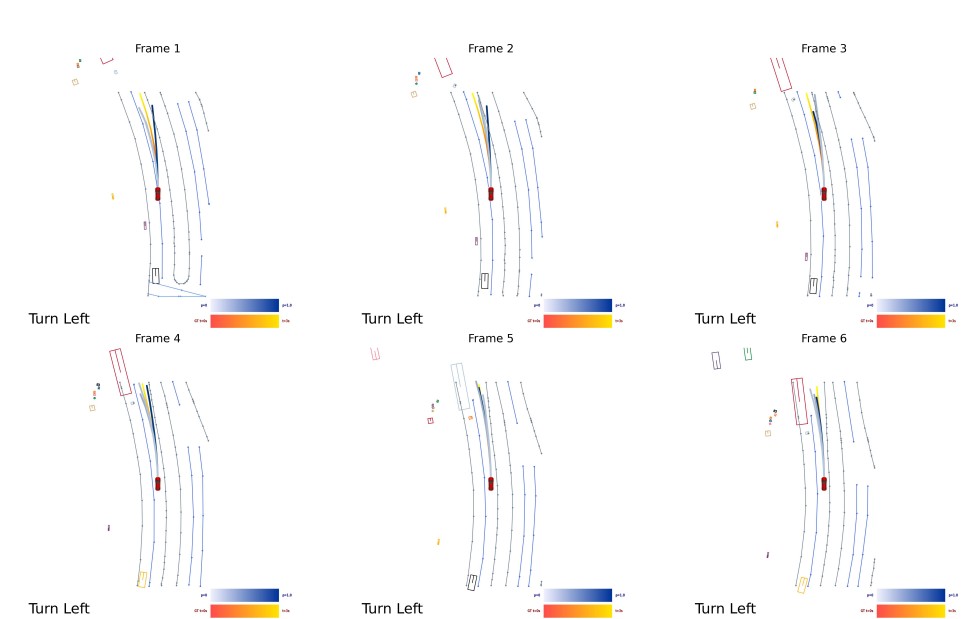

Figure 9: **Qualitative results on the nuScenes dataset.** Our HE-Drive exhibits strong temporal consistency, as shown in consecutive turn left frames.

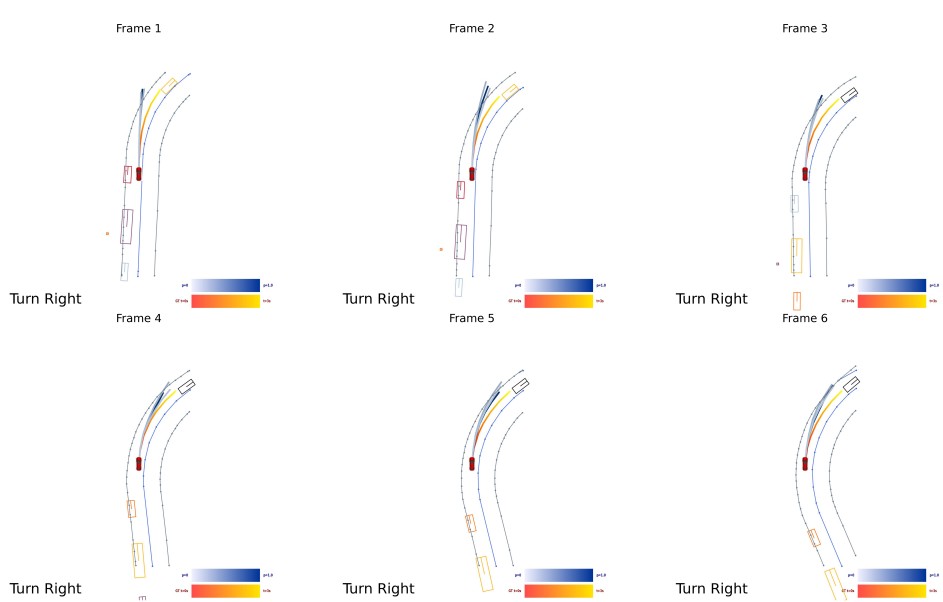

Figure 10: **Qualitative results on the nuScenes dataset.** Our HE-Drive exhibits strong temporal consistency, as shown in consecutive turn right frames.

## B.2 Visual Question Answering (VQA) on nuScenes dataset.

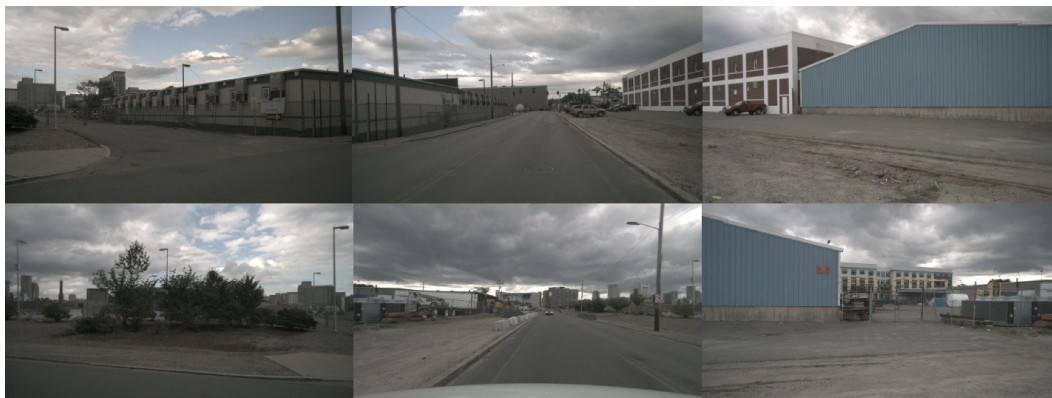

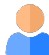

Provide a detailed description of a driving scene from a set of car surround images with 6 perspectives, capturing the critical elements such as *time of day*, *weather conditions*, *road environment*, and *available lane options*.

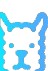

**Time:** Late afternoon.
**Weather:** Overcast and cloudy.
**Driving Scenario:** Urban industrial area with light traffic and visible construction activity.
**Lane Option:** Two-lane street with clearly marked lanes and parked vehicles on the sides.

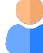

Please list and frame the key objectives in the front view that will influence the next driving decision.

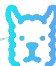

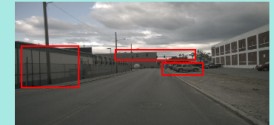 1. **Traffic Lights:** Observe the traffic signals ahead to determine whether to stop or proceed.
2. **Parked Vehicles:** Be aware of vehicles parked along the side, which might require slight steering adjustments.
3.**Building Proximity:** The proximity of buildings may limit side visibility, so remain cautious of any sudden entries onto the road.

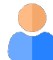

Based on the previous description, should we drive conservatively or aggressively? What level and what score should we use?

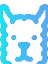

**Driving Style: Aggressive**
**Level**: **II** (where I is the most aggressive)
**Speed Weight: 1.0 | Longitudinal Comfort Weight: 1.0**
Justification:
- Visibility: Daylight provides enough visibility despite the overcast skies.
- Traffic: Light traffic allows more freedom to manoeuvre quickly.
- Obstacles: Awareness of parked cars and potential construction mitigates sudden surprises.

Figure 11: **Qualitative results of the trajectory scorer guided by Llama 3.2V.** We show the questions (Q), context (C), and predicted answers (A) on nuScenes dataset.

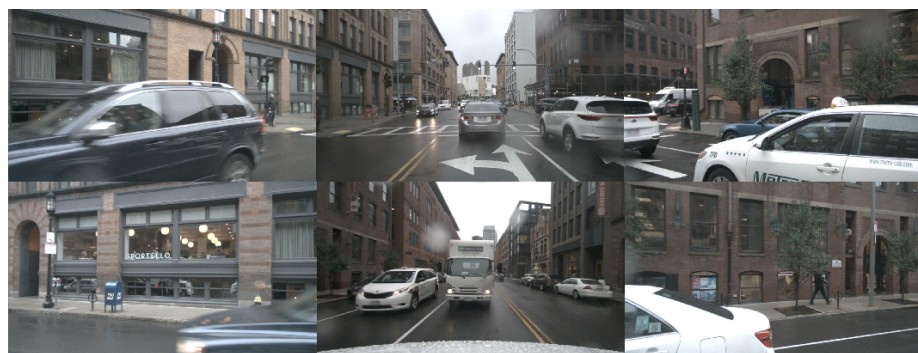

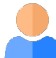

Provide a detailed description of a driving scene from a set of car surround images with 6 perspectives, capturing the critical elements such as *time of day*, *weather conditions*, *road environment*, and *available lane options*.

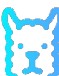

**Time:** Daytime.
**Weather:** Rainy or cloudy, evidenced by water droplets and gloom.
**Driving Scenario:** In an urban area with moderate traffic at an intersection.
**Lane Option:** Two lanes with left and straight directional arrows, allowing choices to either go straight or turn left.

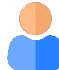

Please list and frame the key objectives in the front view that will influence the next driving decision.

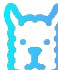

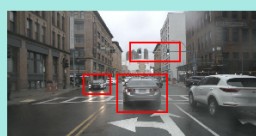

1. **Traffic Lights:** Observe signals affecting your route.
2. **Vehicles ahead:** Observe the movement of the vehicle ahead.
3. **Oncoming vehicles:** When turning left, please pay attention to oncoming vehicles

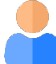

Based on the previous description, should we drive conservatively or aggressively? What level and what score should we use?

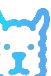

**Driving Style: Conservative**
**Level: I** (highest level of conservatism)
**Speed Weight: 1.5** | **Centripetal Acceleration Weight: 3.5**
Justification:
- Given the urban setting, rainy weather, wet roads, and busy traffic with pedestrians, a conservative driving approach is advisable.

Figure 12: **Qualitative results of the trajectory scorer guided by Llama 3.2V.** We show the questions (Q), context (C), and predicted answers (A) on nuScenes dataset.

### B.3 REAL-WORLD QUALITATIVE RESULTS.

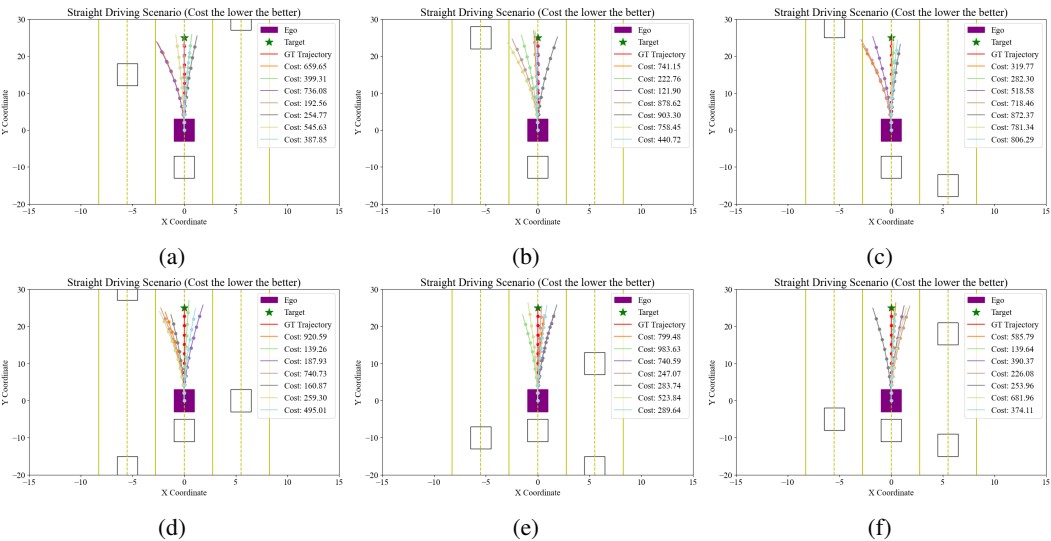

Figure 13: **Qualitative results on real-world datasets.** The subfigures depict the results from the evolution of the trajectory in the real world at different timestamps: 1s, 2s, 3s, 4s, 5s, and 6s.

### B.4 OPENSCENE DATASET VISUALIZATION RESULTS

As shown in Figure 14b, HE-Drive can generate multi-mode trajectories with strong temporal consistency on the closed-loop dataset OpenScene for selecting vehicle control.

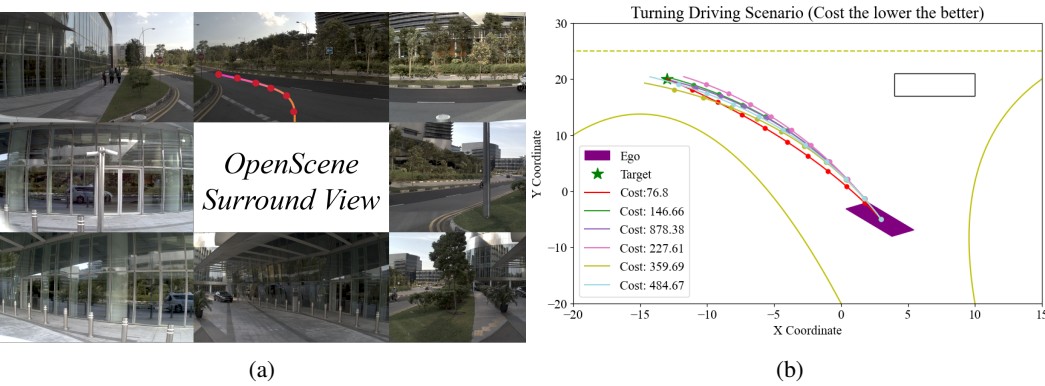

Figure 14: Figures (a) and (b) showcase the trajectory generation and scoring process, with the optimal path, indicated by the red trajectory in (b), being selected for vehicle control based on the lowest cost criterion.

1026
1027
1028
1029
1030
1031
1032
1033
1034
1035
1036
1037
1038
1039
1040
1041
1042
1043
1044
1045
1046
1047
1048
1049
1050
1051
1052
1053
1054
1055
1056
1057
1058
1059
1060
1061
1062
1063
1064
1065
1066
1067
1068
1069
1070
1071
1072
1073
1074
1075
1076
1077
1078
1079

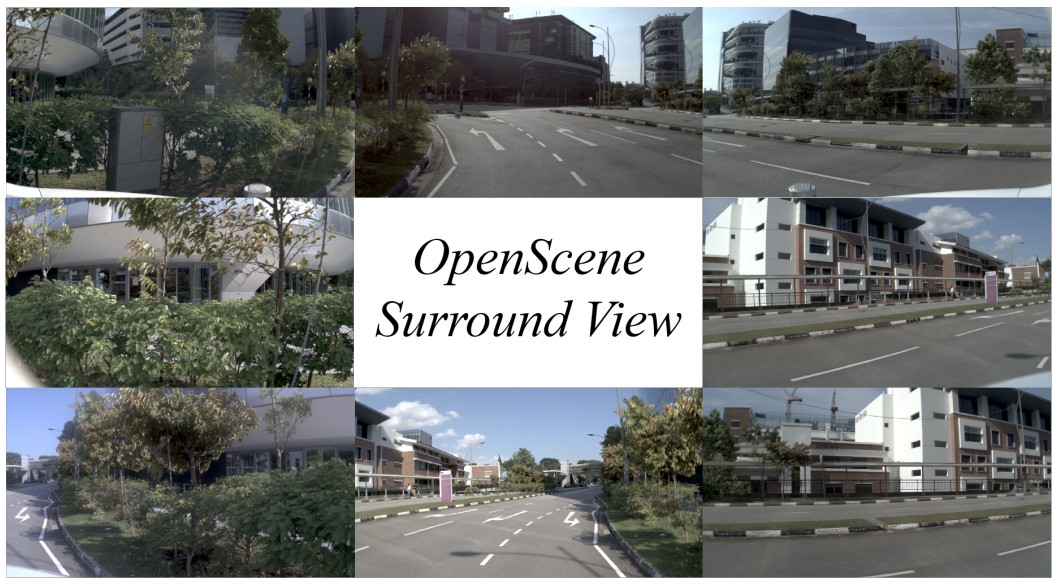

**Q: Provide a detailed description of a driving scene from a set of car surround images with 6 perspectives, capturing the critical elements such as *time of day*, *weather conditions*, *road environment*, and *available lane options*.**

**A:**
**Time of Day and Weather:** The presence of shadows and bright sunlight suggests that it is either late morning or early afternoon. The sky is clear and blue with a few scattered clouds, indicating pleasant weather conditions with no precipitation.
**Road Environment**: The area appears to be part of a city or business district with modern buildings and greenery lining the streets. Trees and bushes are lush and well-maintained, adding to the urban landscaping.
**Buildings and Surroundings:** The buildings are modern, with a mix of brick and glass facades. Their architecture suggests they might be educational or office buildings. Pedestrian walkways are visible, and some individuals are seen walking in the vicinity, indicating moderate foot traffic.
**Lane Options and Road Features**: The road has multiple lanes with clear white markings for traffic direction. Arrows on the road surface indicate available lane options: straight, left, and right turns. The presence of a U-turn sign in one area highlights additional navigational options for drivers. Traffic lights and pedestrian crossing signals are also evident, controlling the intersection.

**Q: Please list and frame the key objectives in the front view that will influence the next driving decision.**

**A:**
**Lane Markings:** Clear lane markings showing the permissible directions (straight, left, and right turns) which will guide the choice of lane and direction.

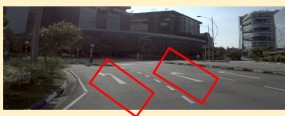

**Q:Based on the previous description, should we drive conservatively or aggressively? What level and what score should we use?**

**A:**
**Driving Style: Aggressive  Level: II** (where I is the most aggressive)  **Speed Weight: 1.0 | Longitudinal Comfort Weight: 1.0**
Justification:  - No pedestrians or vehicles can move fast.
- The presence of buildings, pedestrian traffic, and signalized intersections requires careful attention.
-  Clear lane markings suggest deliberate lane changes and attentive navigation.

Figure 15: **VQA results on OpenScene.**  Llama 3.2V can accurately perceive and identify key targets in the current scene, such as turning representations, and then give the weights that need to be adjusted and finally explain the reasons.

