# OpenReview forum: "HE-Drive: Human-Like End-to-End Driving with Vision Language Models"
_ICLR.cc/2025/Conference — ICLR 2025 Conference Withdrawn Submission_

### Official Review · Reviewer_in1n · 2024-10-27

**Soundness:** 2
**Presentation:** 3
**Contribution:** 2
**Rating:** 5
**Confidence:** 4

**Summary:**

The paper presents an end-to-end autonomous driving system to generate temporal consistent and comfortable trajectories for self-driving vehicles. The system consists of three modules, i.e. a sparse perception module that extracts features from multi-view images, a diffusion model based motion planner that generates multi-modal temporal consistent trajectories, and a VLM enhanced trajectory scorer that chooses the most comfortable trajectory to output. The experimental results conducted on both closed-loop and open-loop benchmarks show the superiority of the proposed system.

**Strengths:**

In general, the paper is well-written and easy to understand. The paper shows the novel application of diffusion models and VLMs in autonomous driving domain. The experimental results are quite promising, indicating the effectiveness of the proposed system.

**Weaknesses:**

The claimed contributions are not well supported by the experiments. Specifically,
1) The authors claim that the diffusion-based motion planner as one of their contributions that can generate temporal consistent trajectories. However, the temporal inconsistency issue is addressed by incorporating the historical predicted trajectories as additional inputs to the motion planner. I don't see the necessity of using a diffusion model. On the other hand, diffusion models are computational intensive especially in the inference time since it requires multiple (hundreds of) iterations for denoising. The authors didn't specify how many iterations they used, but they achieved much higher FPS which is quite counter-intuitive. The authors need to justify the benefits of using a diffusion model as the motion planner.
2) It is also unclear to me if it is necessary to use the VLM for trajectory scorer. The VLM is just used to adjust the weights of the different components in rule-based scorer. Though the ablation study shows that VLM helps to improve the performance, it is not clear how the weights are defined without the VLM. Is there any more efficient way such as training a small scenario classifier to determine the weights, rather than using the heavy VLM and initiating multiple QA sessions to get the answers?
3) The proposed comfort metric is not sound. The comfort metric is defined by calculating the difference between the predicted trajectory and the ground truth trajectory. In this way, the soundness of the metric heavily depends on the quality and comprehensiveness of the collected trajectories which are difficult to fulfill. There might be lots of comfortable trajectories that are far different from the collected ones.

**Questions:**

1) How many iterations are used to denoising the trajectories during the inference time? i.e. What is the parameter k in diffusion-based motion planner? How does the diffusion model meet the  latency criteria and achieve the high FPS?
2) Is it possible for other motion planners, like transformer-based model, to incorporate the historical predictions and address the temporal consistency issue?
3) How does the rule-based scorer determine the weights of different costs?  Is there any more efficient way such as training a small scenario classifier to determine the weights?

---

### Official Review · Reviewer_KyuD · 2024-10-28

**Soundness:** 1
**Presentation:** 2
**Contribution:** 1
**Rating:** 3
**Confidence:** 4

**Summary:**

HE-Drive presents an end-to-end autonomous driving system, which encompasses a sparse perception module, a diffusion-based motion planner, and a trajectory scorer. The system generates 3D spatial representations via sparse perception and utilizes a diffusion-based motion planner to produce temporally consistent multi-modal trajectories. Subsequently, the trajectory scorer selects the most comfortable trajectory. The experimental findings demonstrate that HE-Drive has attained remarkable performance on multiple datasets, such as reducing the collision rate, enhancing efficiency, and improving comfort, thereby showcasing its potential in the autonomous driving field. However, there are also potential issues like limited scene adaptability. And the whole paper makes me feel very engineering-oriented. It doesn't solve the scientific problems faced by the current system. It simply piles up more modules with better performance. Moreover, the application of the Llama model in this work doesn't have a good comparison and ablation. It seems to be used directly, and there is a large amount of prompt work that needs to be tried and improved.

**Strengths:**

1. Put forward the HE - Drive system that integrates multiple modules, combining the advantages of each module, such as sparse perception, diffusion-based motion planning, and trajectory scoring system.
2. Carry out comprehensive experimental verification on multiple datasets to demonstrate the performance of the system.

**Weaknesses:**

1. The paper is too engineering-oriented and fails to solve the core scientific problems of the autonomous driving system. End-to-end Autonomous driving implementation is not about assembling building blocks or piling up seemingly best modules and models. I am truly concerned about the rationale behind choosing specific models.
2. The integration of different modules in the HE - Drive system lacks a comprehensive scientific analysis. While the paper describes how each module functions individually, it fails to explain how they interact and contribute to the overall performance from a scientific perspective.

**Questions:**

1. Module Selection Justification
a. What are the differences between choosing diffusion and other policies like BC?
b. Is there any comparison presented?
c. Why is Llama used?
d. What is the original intention of the scoring model?

2. Module Interaction
a. How does the sparse perception module enhance the performance of the diffusion-based motion planner?
b. How do the outputs of different modules combine to produce a more effective driving trajectory?

3. Module Combination Motivation
a. What is the specific motivation for using this module combination?
b. Are there any alternative combinations that were considered and why were they rejected?

4. Module Role and End-to-End System
a. What is the authors' perspective on each module's role and significance within the system?
b. How is the end - to - end process achieved with the integration of these modules?

Most important:  Are there any scientific problems that you attempt to solve in this work? Do you want to do a framework-type of work, or simply want to use these latest works, such as diffusion or Llama?

**Details Of Ethics Concerns:**

None. There is no ethics review needed.

---

### Official Review · Reviewer_hvXi · 2024-11-03

**Soundness:** 1
**Presentation:** 2
**Contribution:** 2
**Rating:** 3
**Confidence:** 4

**Summary:**

The paper proposes an end-to-end driving method based on a diffusion model for trajectory prediction. The diffusion model can learn multi-modal trajectories and is therefore conceptually superior to standard imitation learning which can only model one mode. In addition this diffusion model gets the past predictions as input to increase the temporal consistency of the predictions. The authors propose to sample several trajectories and choose the best one based on a combination of cost functions. A VLM is used to describe the scene and come up with a driving style decision as well as weights for the cost function. The authors claim to get better driving performance as well as smoother behaviour.

**Strengths:**

S.1. **Diffusion policy:** Using a diffusion model for trajectory prediction is an interesting idea to model multi-modal trajectories.

S.2. **VLM** for interpretable decisions. Using a VLM to obtain decision that can be easy to understand by humans as the interface is natural language.

S.3. The writing and figures are clear and easy to understand (however some key information are missing).

**Weaknesses:**

**W.1 Evaluations**

**W.1.1 NuScenes as planning benchmark**

1. As stated by multiple paper open-loop evaluation and therefore also the NuScenes dataset is not a good benchmark to evaluate planning, as shown by: [1, 2, 3, 4]. Using NAVSIM with the OpenScene dataset (which is still open-loop though → see W2.2 or using closed-loop simulators like CARLA would be the best to obtain trustworthy results.
2. Metrics: As several works [5,6,7] showed, the implementation of the L2 and collision metrics differ for different papers and implementations (especially [7] has a detailed breakdown of the differences). It is not clear which one was used in this paper.

**W.1.2 Correctness of Results on OpenScene Dataset**

1. Closed-Loop benchmark: The authors states that they use an closed-loop evaluation (line. 099+349+509), but NAVSIM is not fully closed-loop.
2. Clearance of description: The authors use the NAVSIM evaluation framework. The description is not clear on the exact setting (it is not mentioned that the NAVSIM framework is used) Would be good to make this clearer. Also since NuScenes planning is not a good benchmark to evaluate planning results (see **W.1.1**) the NAVSIM results would be actually the more trustworthy results. However the implementation of the model and training details are not sufficient. Some open questions: Is the architecture exactly the same as for the NuScenes results? On what data is it trained?
3. Efficiency: It is unclear where the numbers in Figure 6 b) are coming from. In my understanding PDM-closed is a rule-based system and therefore has 0h training time but is reported with 62h training time. Also TransFuser reports that they train for 24h on 1 GPU but it is reported as 28h. HE-Drive in comaprison is trained on 8 GPUs, so it would be good to compare GPU hours and not total hours. Also it would be good if the authors could report where the numbers for FPS and training time are coming from.
4. Missing comparisons: The actual state of-the art models are missing in Table 5. In the Hydra-MDP paper are better numbers reported than what is stated in Table 5. Also TransFuser has a better score in the official paper and leaderboard than what is stated here. Was there a reason for this?
5. Official Leaderboard: It would be good if the authors could submit there model to the official Leaderbaord and report official numbers.

**W.1.3 Smoothness and temporal consistency**

1. Metric: Since the driving comfort metric is newly proposed by the authors it would be good to have qualitative example to showcase how well the metric aligns with human judgement. It is also hard to tell what score in the comfort metric would be an acceptable score by human judgement (i.e. for which score would a human in a car say that the behaviour is smooth enough?).
2. The authors state that it “reduce sharp side-to-side movements, sudden braking” (line 266) does this affect safety-critical scenarios where sudden braking (i.e., emergency brake) is necessary?
3. As this is one of the main motivations in the paper it would be good to have proper ablations on this. The authors claim that “such trajectory planners and scorers face the dilemma of generating *temporally inconsistent* and *uncomfortable* trajectories.” (line 036), and that hey can “generate accurate and temporally consistent trajectories.” (line 359).  How much does the VLM improve the smoothness? Would be a post-processing of the other methods be enough to obtain good comfort performance (e.g. averaging previous predictions?)

**W.1.4 Comparison with fixed scoring weights**

I was wondering how much the fixed weights are tuned and how much performance gain can be achieved with a better tuned set of weights.


**W.2. Related Work**

1. Section 2.1 END-TO-END AUTONOMOUS DRIVING is missing some crucial papers for the closed-loop setting [TransFuser, InterFuser, etc.]
2. 2.2 DIFFUSION MODELS FOR TRAJECTORY GENERATION: is also missing some crucial related work: [8,9, 10, 11]. Also would be nice to know how the method of this paper compares to the related works.

[1] Dauner, Daniel, et al. "Navsim: Data-driven non-reactive autonomous vehicle simulation and benchmarking." Neurips (2024).

[2] Li, Zhiqi, et al. "Is ego status all you need for open-loop end-to-end autonomous driving?." *Proceedings of the IEEE/CVF Conference on Computer Vision and Pattern Recognition*. 2024.

[3] Zhai, Jiang-Tian, et al. "Rethinking the open-loop evaluation of end-to-end autonomous driving in nuscenes." *arXiv preprint arXiv:2305.10430* (2023).

[4] Codevilla, Felipe, et al. "On offline evaluation of vision-based driving models." *Proceedings of the European Conference on Computer Vision (ECCV)*. 2018.

[5] Mao, J., Ye, J., Qian, Y., Pavone, M., & Wang, Y. (2023). A language agent for autonomous driving. *arXiv 2023*

[6] Mao, J., Qian, Y., Zhao, H., & Wang, Y. (2023). Gpt-driver: Learning to drive with gpt. *arXiv 2023*

[7] Weng, Xinshuo, et al. "PARA-Drive: Parallelized Architecture for Real-time Autonomous Driving." *Proceedings of the IEEE/CVF Conference on Computer Vision and Pattern Recognition*. 2024.

[8] Chi, Cheng, et al. "Diffusion policy: Visuomotor policy learning via action diffusion." *The International Journal of Robotics Research* (2023)

[9] Pearce, Tim, et al. "Imitating Human Behaviour with Diffusion Models." *The Eleventh International Conference on Learning Representations (ICLR 2023)*. 2023.

[10] Reuss, Moritz, et al. "Goal-conditioned imitation learning using score-based diffusion policies." Robotics: Science and Systems, 2023

[11] Janner, Michael, et al. "Planning with Diffusion for Flexible Behavior Synthesis." *International Conference on Machine Learning*. PMLR, 2022.

**Questions:**

1. My biggest concern is the evaluation of the proposed method. Having the ablations on NAVSIM (or even closed-loop results) with a fair comparison with related work would add a lot.

2. Having misleading statements about closed-loop evaluation (OpenScene with NAVSIM ist not closed-loop but a non-reactive open-loop simulation), missing state-of-the-art results in Table 5 and unclear origins of the results in Figure 6b) makes it hard to trust the results. It would be highly appreciated if the authors could add the missing descriptions and fix the misleading ones for more clarity.

---

### Official Review · Reviewer_dysz · 2024-11-03

**Soundness:** 3
**Presentation:** 3
**Contribution:** 3
**Rating:** 6
**Confidence:** 4

**Summary:**

This paper has two contributions, a diffusion-based motion planner and a vision language model guided trajectory scorer. The authors build on a previous work called SparseDrive to utilize a sparse representation of the driving scene. This scene representation is then used for conditioning the diffusion model along with the ego status and past trajectories. The sampled trajectories are then scored using a weighted average of safety and comfort scores. The VLM is periodically invoked to adjust the weights. The authors produce comparison of open and closed loop driving performance with other state of the art models.

**Strengths:**

The results of the paper are very good, both in terms of driving performance and speed. Especially the comparison in closed loop environment is very competitive.
The diffusion-based trajectory generation is done in a novel way, with consideration to maintaining temporal consistency.

**Weaknesses:**

There is relatively less details on the sparse perception part in this paper. It is understood that this work is an extention of the SparseDrive work, however more details on how this sparse scene representation is used in the diffusion model would have been helpful.
The results in the closed loop section (table 5) should have included all the comparisons shown in the graphs as well.
The motivation for VQA task is unclear, it seems the paper only uses the VLM for updating the weights of the trajectory score function in every 5 seconds. However, what happens to the dialog, it's not clear if it is validated or can be used in any way later.
There are quite a few instances of spelling, grammar mistakes and typos.

**Questions:**

There are several topics that are unclear in this paper:
How is data curation done for the stage one of the VLM guided trajectory scorer?
How do you initialize the weights of the trajectory scoring metric?
Why do you pick 5 sec as the calling frequency for Llama?
How did you define the aggresive and conservative driving style parameters? Did you validate with human data?
Is the VLM also adjusting parameters beyond the profiles of aggresive and conservative driving? Is there fine grained control of speed / comfort weights?
How do you claim these results are better / as good as hyper-parameter optimization of the weights? Did you perform such a study? (table 3?)
What are HE-Drive-S and HE-Drive-B? They need to be described clearly.
What do you do with the VQA? How can this conversation be used for driving? Did you perform any quantitative analysis on the results?

---

### Official Review · Reviewer_3RUQ · 2024-11-04

**Soundness:** 2
**Presentation:** 1
**Contribution:** 2
**Rating:** 3
**Confidence:** 4

**Summary:**

The paper proposes HE-Drive, an end-to-end autonomous driving system designed to tackle the issues of temporal inconsistency and discomfort in trajectories. To address these issues, the system leverages a sparse perception pipeline that generates 3D spatial representations. These representations serve as inputs for a Conditional Denoising Diffusion Probabilistic Model-based motion planner, producing multi-modal trajectories. Additionally, a Vision-Language Model (VLM)-guided trajectory scorer selects the best trajectory.

Experiments are conducted on datasets like nuScenes, a real-world dataset, and OpenScene to evaluate the performance and efficiency. Experimental results have been strong, such as achieving a 71% reduction in average collision rate compared to VAD and operating 1.9× faster than SparseDrive.

**Strengths:**

1. **Insightful Use of Diffusion Models and VLMs**: The authors’ recognition that offline expert trajectories suffer from out-of-distribution (OOD) issues and generalization challenges underpins their choice of a diffusion model for trajectory generation. Besides, utilizing Vision-Language Models (VLMs) as a trajectory scorer rather than as a trajectory predictor is a unique approach that diverges from traditional methods. This design decision potentially enhances robustness and aligns with the paper's objective of improving comfort in trajectory planning.

2. **Comprehensive Performance Evaluation**: The model demonstrates strong performance and efficiency relative to prior Camera-based and LiDAR-based approaches. The inclusion of well-structured ablation studies is valuable, as it offers a thorough understanding of how individual components contribute to the model’s overall success.

**Weaknesses:**

1. **Overclaim on “Human-like” Design**: The paper repeatedly emphasizes “human-like” characteristics, claiming HE-Drive to be the “first human-like end-to-end driving system.” However, the term “human-like” lacks clear definition and justification. Human cognitive processes are complex, and many algorithms in autonomous driving draw inspiration from these processes, such as top-ranking methods on CARLA leaderboard [3,7]. Thus, it’s unclear in what specific ways this work is pioneering in achieving “human-like” behavior, and additional clarity is needed to support this claim. Specifically, authors should provide a clear definition of what they mean by "human-like" in the context of their system, and specify how their approach differs from or improves upon existing methods that also aim for human-like driving behavior.

2. **Lack of Novelty and Ambiguous Contribution**: Although the paper frames temporal inconsistency and uncomfortable trajectories as key issues, motion comfort has been well-studied in the motion planning and control community [1,2], and the paper’s approach—introducing a comfort metric and incorporating previously predicted trajectories as additional inputs—feels somewhat basic, trivial, and lacks novelty. The authors should explain what novel aspects their comfort metric and use of previously predicted trajectories bring to the field. Otherwise, the authors would benefit from revisiting the paper and figuring out the true novelty that are unique contributions to these areas.

3. **Limited Coverage in Literature Review**: In the introduction, the authors seem to assume that all end-to-end methods use scoring mechanisms for trajectory generation. However, dominant end-to-end methods also generate trajectories through motion optimization [3,4] and network prediction [5,6]. Expanding the discussion to include these approaches would provide a more balanced perspective.

4. **Lack of Clarity on Dialogue Mechanism for Hallucination Mitigation**: In line 296, the authors mention that dialogues can mitigate hallucinations, but they do not specify the type of dialogues used or explain why this approach effectively reduces hallucinations. A clearer explanation would be necessary.

5. **Insufficient Explanation of Method Choices**: The choice of a diffusion model and VLMs for addressing temporal inconsistency requires further explanation. While in the introduction section, the authors mention that the proposed method aims to address the issue of temporal correlation and generalization, they do not clearly demonstrate how the proposed method tackles these challenges in later sections. Additional reasoning behind this methodological choice would enhance the paper’s technical soundness.

6. **Unexplained Terminology and Method Details:**
   - Line 74: The term “lifelong evaluation” lacks clarification, what does it mean?
   - Line 161: The specifics of the “compact 3D representation” are unclear; is it using 3D voxels, sparse voxels, or tokens?
   - Line 196 and Figure 2: The term “observation” is used ambiguously. Given that the model already uses 3D representations as an environmental proxy, it’s unclear what the “observation” specifically refers to.

7. **Experimentation Gaps:**
   - Introduction to Baselines: A brief description of the major compared methods would enhance the reader’s understanding of their methodologies, strengths, and limitations relative to the proposed method.
   - Real-world Dataset Information: Three datasets are used (nuScenes, OpenScenes, and a real-world dataset), but the authors did not introduce the real-world dataset in sufficient detail. Specifically, they should provide information on the dataset’s name, its origin (e.g., collected by a specific institution or organization), and the volume of data it contains. Including these details would give readers a clearer understanding of the data diversity and scope, allowing for a more comprehensive assessment
   - Evaluation of temporal consistency: In Section 4.2, the authors show some qualitative illustrations to demonstrate the temporal consistency of the model.  However, a quantitative evaluation of temporal consistency would be necessary to provide an objective measure of the model’s performance in this area.
   - Trajectory Scorer Methodology: The first row of Table 3 suggests that a rule-based and a VLM-based scorer are used simultaneously, which is confusing. Clarification is needed, such as whether these scorers are used simultaneously or in some sequential manner, and how their outputs are combined if both are used.
   - Another ablation: Adding another ablation, which replaces the diffusion model with a fixed trajectory set could offer additional insights into the benefits and drawbacks of diffusion models for generating trajectory candidates.

7. **Presentation Issues**:
   - Section 3.3 Placement: The paragraph beginning on line 216 may be more appropriately located under Section 3.3 instead of Section 3.2.
   - Figure Legends: Figures 9 and 10 have legends that are too small to read comfortably.
   - Demo in Main Text: Including a brief demonstration (as in the appendix) of the proposed method within the main text would assist readers in understanding the workflow and methodology of HE-Drive more effectively.

[1] Gu, T., 2017. Improved trajectory planning for on-road self-driving vehicles via combined graph search, optimization & topology analysis (Doctoral dissertation, Carnegie Mellon University).

[2] Wang, L., Sun, L., Tomizuka, M. and Zhan, W., 2021. Socially-compatible behavior design of autonomous vehicles with verification on real human data. IEEE Robotics and Automation Letters, 6(2), pp.3421-3428.

[3] Shao, H., Wang, L., Chen, R., Li, H. and Liu, Y., 2023, March. Safety-enhanced autonomous driving using interpretable sensor fusion transformer. In Conference on Robot Learning (pp. 726-737). PMLR.

[4] Hu, Yihan, et al. "Planning-oriented autonomous driving." Proceedings of the IEEE/CVF Conference on Computer Vision and Pattern Recognition. 2023.

[5] Mao, J., Qian, Y., Ye, J., Zhao, H. and Wang, Y., 2023. Gpt-driver: Learning to drive with gpt. arXiv preprint arXiv:2310.01415.

[6] Chitta, K., Prakash, A., Jaeger, B., Yu, Z., Renz, K. and Geiger, A., 2022. Transfuser: Imitation with transformer-based sensor fusion for autonomous driving. IEEE Transactions on Pattern Analysis and Machine Intelligence, 45(11), pp.12878-12895.

[7] Shao, H., Wang, L., Chen, R., Waslander, S. L., Li, H., & Liu, Y. (2023). Reasonnet: End-to-end driving with temporal and global reasoning. In Proceedings of the IEEE/CVF conference on computer vision and pattern recognition (pp. 13723-13733).

**Questions:**

see the weakness section

---

### Note · Authors · 2024-11-13

I have read and agree with the venue's withdrawal policy on behalf of myself and my co-authors.